# A Review of Critical State Joint Estimation Methods of Lithium-Ion Batteries in Electric Vehicles

**Junjian Hou, Tong Li, Fang Zhou \*** **, Dengfeng Zhao, Yudong Zhong, Lei Yao** and **Li Zeng**

Mechanical and Electrical Engineering Institute, Zhengzhou University of Light Industry, Zhengzhou 450000, China
* Correspondence: 2020021@zzuli.edu.cn

**Abstract:** Battery state of charge (SOC), state of health (SOH), and state of power (SOP) are decisive factors that influence the energy-management system (EMS) performance of electric vehicles. However, the accurate estimation of SOC, SOH, and SOP remains a challenge due to the high nonlinearity of the battery dynamic characteristics and the strong coupling among the states. In this paper, different methods of single-state and two-state joint estimation are classified and discussed, including SOC/SOH and SOC/SOP joint estimation methods, and their advantages and limitations are analyzed. On this basis, key issues of joint multi-state estimation are discussed, and suggestions for future work are made.

**Keywords:** electric vehicle; Lithium-ion battery; core state; joint estimation; fusion technology

## 1. Introduction

A battery-management system (BMS) is a product or technology that manages and controls a power battery in some way [1]. Its main task is to provide the status information required for energy management and vehicle control, which ensure the safety and reliability of the power battery system [2,3]. The basis of energy management and control is to accurately and efficiently monitor the state information of the power battery, including the state of charge (SOC), state of health (SOH), and state of power (SOP) of the battery, etc. [4]. The locations of SOC, SOH, and SOP in the BMS are shown in the Figure 1.

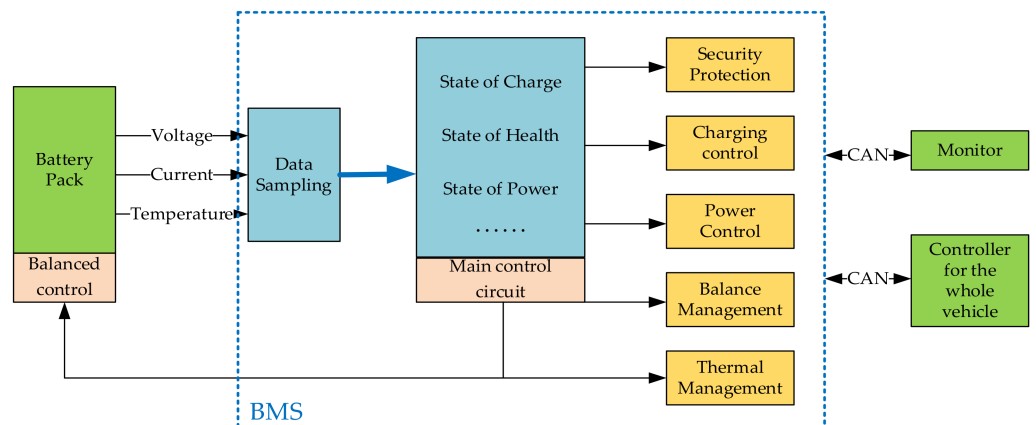

**Figure 1.** BMS architecture diagram.

Obviously, SOC, SOH, and SOP estimation is the key function in BMS, and much work has been completed in this field of research. Many traditional methods of SOC estimation have been developed, such as the ampere–hour counting method, open circuit voltage method, and alternating current (AC) impedance method. These methods establish the

correspondence between the external parameters of power battery (voltage, resistance) and the SOC to achieve the estimation of the power battery SOC with the disadvantage of being time-consuming and unable to be implemented online [5–8]. Model-based methods use an equivalent circuit to model battery dynamic characteristics and achieve the SOC estimation during the iterative operation of filtering methods, such as the Kalman filter (KF). Their disadvantages focus on the high dependence on the accuracy of the model parameters, which are very sensitive to aging, temperature, and SOC of the battery [4]. Their advantages are ease of engineering application and ease of understanding [9,10]. Data-driven methods estimate states with neural network model, which are trained by big data. Their advantages lie in high precision, and the disadvantages include the high uncertainty and neglect of the chemical characteristics of the battery affected by the temperature [11–14]. Similarly, the SOH-estimation method includes model-based and data-driven methods, the strengths and shortcomings of which are similar to the SOC-estimation method. As the resistance and capacity are affected by SOC, the value of SOH is affected by SOC [15,16]. The multi-constraint dynamic method is a common method of SOP estimation, which integrates multiple constraint variables (terminal voltage, current, SOC, etc.) to predict power-battery SOP in real time, and these variables change in real time with the battery SOH. As a result, the SOP estimation will not be credible if the influence of battery SOH and SOC is not considered [14,17–19]. The battery SOC, SOH, and SOP do not exist separately but are coupled with each other, for example, temperature will affect SOC, SOH, and SOP at the same time, and the three will, in turn, cause temperature fluctuations [17]. The SOC estimation needs to consider the influence of SOH on parameters, such as the resistance and capacity used for SOC estimation. The capacity and internal resistance parameters used for SOH are limited by the accuracy of the SOC estimation, and the multi-constrained current estimates used for SOP are obtained under the constraints of SOC- and SOH-related parameters [20]. These show that it is unreasonable to estimate any one of the three states separately without considering the other two state values; furthermore, the development of joint estimation methods for the coupled states is essential to achieve a higher estimation accuracy for each component [21]. Retrieved relevant reviews about state estimation on the web of science. It is shown in Table 1.

**Table 1.** Reviews about state estimation on the web of Science.

| State | References |
|---|---|
| Single-state | [6,11,15,16] |
| Dual-state | [17,19,20] |

The reviews in the Table 1 describe the development process of the methods of single-state estimation or dual-state estimation. However, single-state estimation is not compared with dual-state estimation, and the advantages of dual-state estimation are not highlighted. This paper will compare single-state estimation and dual-state estimation and more intuitively explain the advantages of joint estimation.

The advantage of the joint estimation method is that it can improve estimation accuracy and reduce the calculation cost, thus improving the power and safety and reducing costs of the electric vehicle. The use of joint estimation is conducive to promoting the popularity of electric vehicles, in line with the national demand for the development of electric vehicles [1,22,23]. This paper mainly summarizes the joint estimation methods of SOC/SOH and SOC/SOP and proposes the inadequacy of existing research and future prospects.

The structure of this paper is as follows. Section 2 provides a brief description of three common methods for state estimation of SOC, SOH, and SOP, and analyzes the limitations of existing single-state estimation methods: modeling, parameter identification, and experimental validation, which, under the set specific operating conditions, ignores the influence caused by the coupling relationship and thus cannot circumvent the errors [24]. Section 3 analyzes and summarizes the joint SOC/SOH-estimation method and the joint

SOC/SOP-estimation method in recent years. The main logic is to use multiple algorithms to develop parameter and SOC estimation models and interact with each other; furthermore, it develops a SOP estimation model based on the interacted SOC/SOH estimation models. The accuracy of this was significantly improved by joint estimation. Section 4 summarizes the development status and bottlenecks of multi-state joint estimation technology by comparing methods, technical points, difficulties, advantages, and disadvantages, and the future development direction of joint estimation is proposed according to our own understanding to promote the development and promotion of electric vehicles. The structure of this paper is shown in Figure 2.

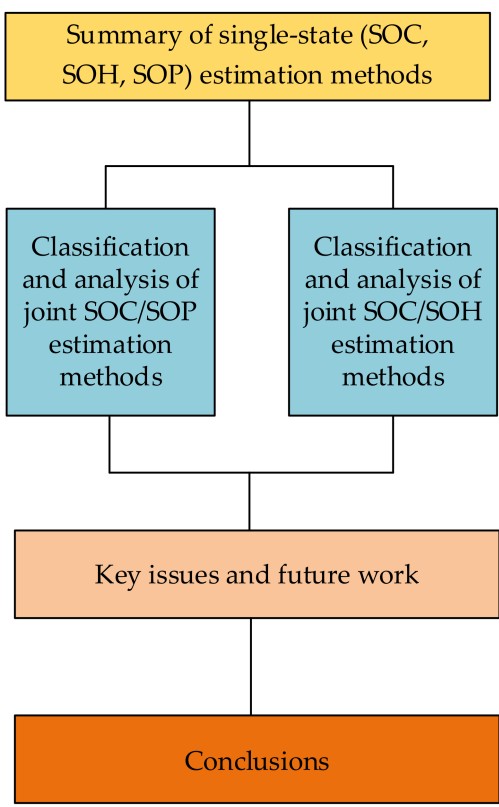

**Figure 2.** The structure of this paper.

## 2. Single-State Estimation

### 2.1. Definition and Estimation Methods of SOC

#### 2.1.1. Definition of SOC

The state of charge of a lithium-ion battery is defined as the percentage of the residual capacity $Q_{current}$ in its maximum available capacity $Q_{rate}$ [10], which is shown as follows:

$$SOC = \frac{Q_{current}}{Q_{rate}} \times 100\% \tag{1}$$

#### 2.1.2. Estimation Methods of SOC

An accurate estimation of SOC is the core technology used to guarantee the rational application of electric energy storage and electric vehicles [25]. Insufficient accuracy in SOC estimation may lead to the overcharging or over-discharging of the battery, thus shortening the battery life or even causing spontaneous combustion, which is harmful to the driver [19]. Furthermore, the control system will not make full use of the energy in the battery pack, resulting in a redundancy of power in a part of the battery, reducing the power output and driving range and increasing the overall vehicle quality and manufacturing cost, which is unfavorable to the promotion of electric vehicles [26]. A large number of studies have

been conducted, and various methods have been proposed to achieve different degrees of improvement in SOC estimation accuracy [27]. A classification of the SOC-estimation methods is shown in Figure 3, and their advantages and disadvantages are summarized in Table 2.

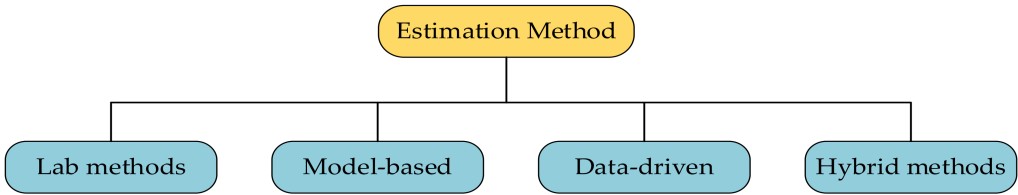

**Figure 3.** SOC-estimation methods classification.

**Table 2.** Advantages and disadvantages of SOC-estimation methods for Li-ion batteries.

| Estimation Methods | Types | Advantages | Disadvantages | Estimation Error |
|---|---|---|---|---|
| Experimental based [23–25] | Ampere-time counting. Open-circuit voltage. AC impedance | Simple principle. Reliable. | Time-consuming. Cannot be estimated in real time. Cumulative error exists | <5% |
| Model-based approaches [19,27,28] | Kalman filter. Particle filter. Sliding mode observer. | Closed-loop estimation. Low requirement for initial SOC values. | Difficult modeling. Difficult parameter identification. | <5% |
| Data-driven approach [21,27] | Neural network class. Support vector machine. Fuzzy logic. | No modeling is required. | High level of data dependency. Time-consuming. | <1.5% |
| Hybrid methods [28–32] | Data-model parallel mixture estimation. Data-model nested mixture model. | High estimation accuracy. Good robustness | Complex calculations. High energy consumption. Slow estimation speed. | <1% |

For SOC estimation, the traditional methods are simple and reliable but make it difficult to accomplish real-time estimations [23–25]. The ampere–time counting method is, by far, the most extensively used method in the traditional methods. In this method, the SOC is estimated by measuring the discharging currents of a battery and integrating them over time. The SOC is calculated by the following equation.

$$SOC(t) = SOC_0(t_0) - \frac{\eta}{C_n} \int_{t_0}^{t} I(t)dt, \tag{2}$$

where, $SOC(t_0)$ is the initial state of charge, $\eta$ denotes the coulombic efficiency, $C_n$ represents rated capacity, and $I(t)$ is the instantaneous discharge current of the battery.

The ECM-model-based SOC evaluation requires the derivation of the circuit models consisting of various circuit elements arranged in series or parallel combinations, such that they replicate the dynamics of the battery. Various ECM models have been proposed, including the Rint model, the RC model, and the Thevenin model [24]. The Thevenin model is used as typical ECM, which is designed using one RC group, a resistance, and voltage source, as depicted in Figure 4.

The model-based estimation method solves the problem of traditional methods, i.e., that they cannot be estimate in real time, and has the advantages of fast estimation and a scientifically rigorous design process. However, its accuracy is easily affected by the modeling accuracy, and its parameters are time-varying in the application process, resulting in nonnegligible model errors [19,27]. In order to avoid the impact of modeling accuracy, a data-driven class method without modeling is proposed. The data-driven estimation

techniques can estimate SOC accurately by measuring battery parameters, including current, voltage, and temperature, thus, battery model, added filter used in model-based approaches can be avoided. Moreover, the network parameters of data-driven methods are determined by the self-learning algorithm. The process is completely different from model-based estimation techniques, where human expertise and substantial time are needed for parameter estimation. The data-driven approaches often require the use of a machine learning (ML) platform in order to obtain the relationship and rules from the data. The basic method of the data-driven method is a neural network (NN). The basic structure of a neural network (NN) consists of a three-layer formation, as shown in Figure 5.

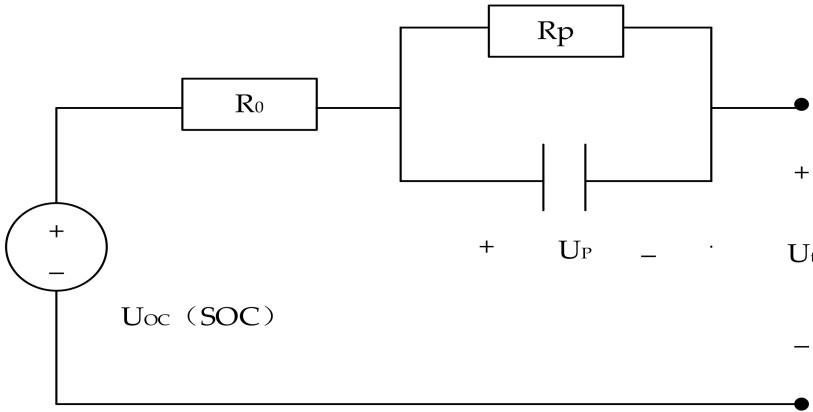

**Figure 4.** The schematic diagram of Thevenin equivalent circuit of LIB.

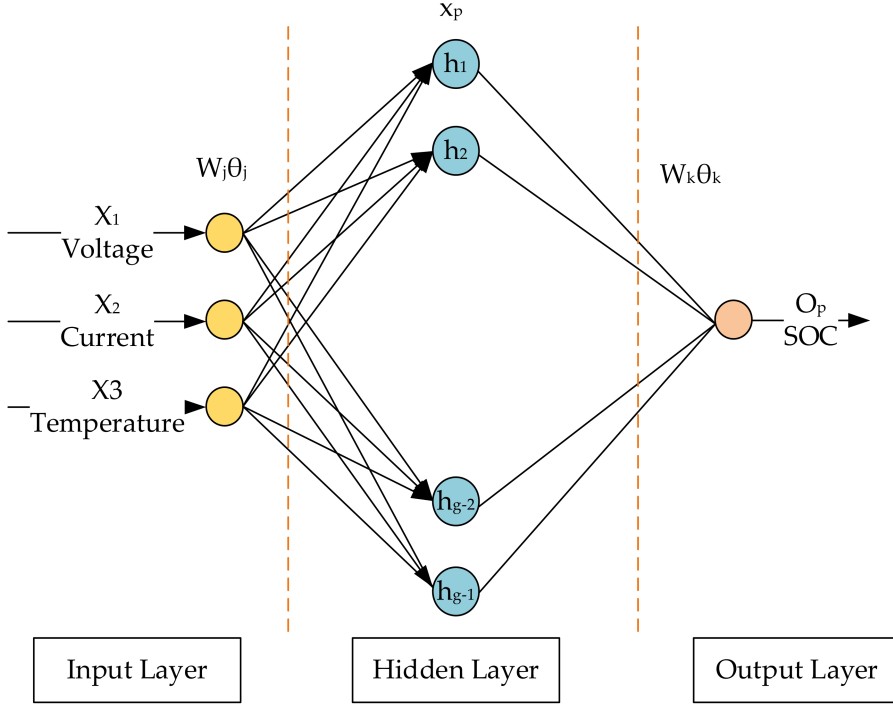

**Figure 5.** The general architecture of the three-layer neural network for SOC estimation.

The input layer takes the vectors of the instantaneous current, voltage, and temperature values. The output layer is the instantaneous SOC value. By training the NN with the input–output pairs, it is able to form a non-linear map that accurately models the input–output relationship without any prior knowledge of the internal structure of the battery. The relationship between the input layer and the output layer is developed using a suitable

number of hidden layers, hidden neurons, and activation functions. The SOC in the output layer can be expressed by:

$$SOC_i = f_i \left\{ \sum_k W_{j,k} O_j + \theta_{j,k} \right\},$$

(3)

where $W_{j,k}$ and $\theta_{j,k}$ denote the weight and bias from the hidden layer to the output layer, respectively, $O_j$ is the output of the hidden layer, and $f_i$ represents the activation function.

The model-based estimation method only relies on the mapping relationship between system inputs and outputs to develop SOC prediction models. Data-driven methods avoid the errors caused by time-varying parameters, but the quality of training data has a huge impact on the accuracy of SOC estimation and has high computational cost and long computation time [21,27]. A hybrid-driven approach is proposed to combine the rapidity of model method estimation with the characteristics of the data-driven nonlinear modeling capability [28]. Hybrid methods are used to improve the accuracy and robustness of SOC estimation. Usually, two or three algorithms are combined to develop a hybrid method. In most cases, the optimization method is employed with model-based and data-driven methods to examine SOC which not only enhances the performance, but also delivers accurate results [29–32].

The authors in [29] proposed a genetic algorithm (GA) to find the optimal battery parameters of ECM in order to estimate SOC using a hybrid pulse power characterization (HPPC) experiment, as shown in Figure 6. A series of actions, including crossover, mutation, and selection, is employed to identify the model parameters. The measured current and battery terminal voltage are assigned as the input and output of the model, respectively, during the process of parameter identification. The fitness value is determined by calculating the difference between the measured voltage values and the model output. The proposed method can estimate the SOC of the LIB pack accurately and prevent the battery pack from overcharging and over-discharging, with the SOC error being less than 1%. The experiment's results also confirm the suitability of the proposed algorithm for online BMS execution.

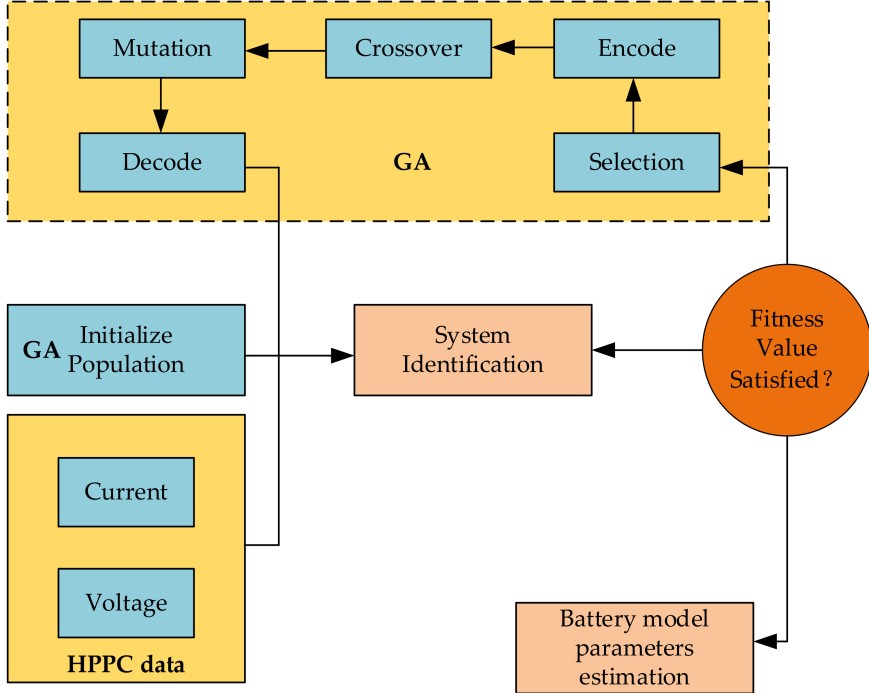

**Figure 6.** Battery model parameters determination using GA.

The SOC-estimation method mentioned above is performed under certain constraints, such as specific temperature conditions, estimating SOC at the cell level, or ignoring the physical and chemical characteristics of the battery. Battery charging and discharging not only changes the SOC, but also causes battery aging, which in turn affects the SOC by affecting the battery capacity. SOP estimation is performed under the SOC constraint, and the heat generated in the cell causes a change in the battery temperature, resulting in parameter variation during SOC estimation. Considering the relationships among SOC and SOH and SOP, ignoring the influence of SOH and SOP, to estimate SOC alone will have inherent defects and limit the estimation accuracy; the development of a joint estimation method for SOC, SOH and SOP is highly necessary.

### 2.2. Definition and Estimation Methods of SOH
#### 2.2.1. Definition of SOH

With the increase of battery charging and discharging times and the accumulation of sheltering time, the battery health status gradually deteriorates, its power and capacity show varying degrees of attenuation, the battery capacity decreases, and the internal resistance increases. Thus, the capacity and internal resistance are commonly used to define SOH.

1. SOH is one of the important parameters of lithium-ion batteries, which is calibrated according to the change of battery capacity, as follows.

$$SOH = \frac{Q_m}{Q_r} \times 100\%, \tag{4}$$

where $Q_r$ is the rated capacity, and $Q_m$ is the current maximum available capacity of the battery, which is measured under rated conditions.

2. SOH is defined according to the internal resistance of the battery, as follows.

$$SOH = \frac{R_e - R}{R_e - R_n} \times 100\%, \tag{5}$$

where $R$ is the internal resistance under the current state, $R_e$ is the internal resistance of the battery when it reaches the end of life, and $R_n$ is the internal resistance of the new battery.

#### 2.2.2. Estimation Methods of SOH

Inaccurate estimation of battery SOH can affect battery life, safety, and reliability [31]. By monitoring the battery SOH in real time to correct the SOC, the operation of the BMS facilitates, and thus improves, the driving safety of electric vehicles, reducing the maintenance and use costs of electric vehicles [32]. Scholars have completed a lot of research on improving the accuracy of SOH estimation; a classification of the methods used for SOH estimation is shown in Figure 7, and their advantages and disadvantages are summarized in Table 3.

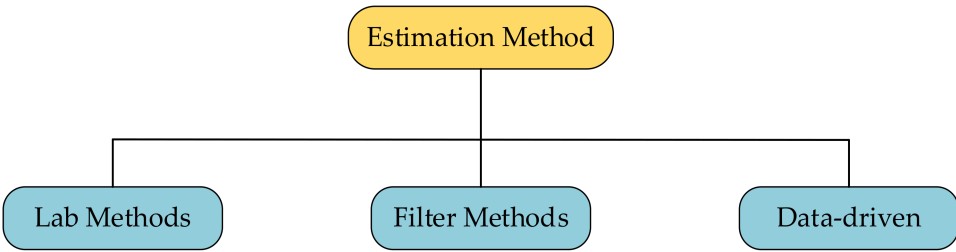

**Figure 7.** SOH-estimation methods classification.

**Table 3.** Advantages and disadvantages of SOH-estimation methods for Li-ion batteries.

| Estimation Methods | Types | Advantages | Disadvantages | Estimation Error |
|---|---|---|---|---|
| Traditional estimation methods [31–34] | Capacity measurement. Resistance measurement. Differential analysis. | Simple principle. Small calculation volume. Accurate. Reliable. | Time consuming. Cannot be estimated in real time. Cumulative error exists. | <5% |
| Methods for filtering classe [32–36] | Kalman filter. Particle filter. Least squares. | Can be estimated in real time. High precision. Good robustness. | Difficult modeling. Difficult parameter identification. Large calculation volume. | <5% |
| Data Driven [35–41] | Support vector machine. Convolutional neural network. | Real-time estimation. Highly adaptive. High accuracy. | High reliance on data accuracy. Time-consuming offline training. | <1% |

The traditional estimation method is simple and efficient. Its disadvantage is that it requires a high experimental environment, and, using this method, it is difficult to accomplish real-time estimation [32]. The adaptive filtering method is suitable for online real-time estimation, which is superior to the laboratory method, but the accuracy of this method relies on the model accuracy and the type of filtering [34]. The data-driven class estimation method does not have to consider model accuracy and filter type, which conquers the shortcomings of the filtering method estimation, but the accuracy of the data-driven class method is highly dependent on high-quality data [41].

The SOH estimation method is also performed under certain constraints, such as defined temperature conditions and cell types [42]. The temperature of the battery is affected by the discharge rate, and the discharge rate is influenced by the SOC and SOP. Therefore, the SOH is affected by the battery SOC and SOP, and the accuracy of SOH estimation alone is limited, and for these reasons the joint estimation of multiple states is necessary.

*2.3. Definition and Estimation Methods of SOP*

The state of power refers to the maximum power that can be continuously used to charge or discharge. Insufficient SOP estimation may lead to a hindered power output of the vehicle, an interrupted power output, and a fluctuating torque, which affects the driving experience [3]. The accurate estimation of SOP during acceleration, regenerative braking, and gradient climbing can ensure the safety of the battery, improve the safety and driving experience of electric vehicles, meet the user's demand on vehicle performance, enhance the audience of electric vehicles, and contribute to the promotion and popularity of electric vehicles [14]. The existing SOP estimation techniques are classified in Figure 8 and their advantages and disadvantages are summarized in Table 4.

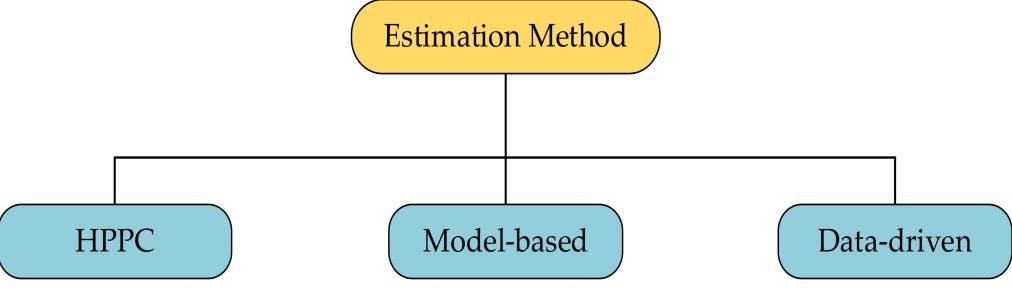

**Figure 8.** SOP-estimation methods classification.



**Table 4.** Advantages and disadvantages of SOP-estimation methods for Li-ion batteries.

| Estimation Methods | Types | Advantages | Disadvantages | Estimation Error |
|---|---|---|---|---|
| Interpolation method [14] | HPPC. | Estimation method is simple. | Requires extensive testing. No consideration of polarization and aging phenomena. | About 3% |
| Model estimation methods [42,43] | Voltage constraint. SOC constraint. Multi-constraint dynamic method. | Simple. Efficient. | Single-state estimation methods have large errors, leading to threats to the safety of the battery. | <5% |
| Data Driven [44,45] | BP neural networks. Adaptive fuzzy neural. Support vector machines. | Good self-learning ability. High accuracy. Good robustness. | Requires extensive experiments. Computationally complex and time-consuming to train offline. | <2.5% |

In SOP estimation, the interpolation method is simple and easy but not suitable for the continuous SOP estimation [45]. To conquer the weakness of the interpolation method, the model estimation method is proposed. Multi-parameter constraints are introduced in the estimation to achieve the prediction of sustained peak power, but the single constraint estimation error is large, and there is a local optimal solution in the multi-constraint estimation, so a data-driven approach is proposed to estimate the SOP. The data-driven estimation method effectively overcomes the local optimal solution problem, with the disadvantage that it requires a large amount of experimental data and is computationally intensive [44]. SOP estimation often uses SOC as a constraint and is influenced by capacity and internal resistance variation caused by battery aging [44]. From the above analysis, it is clear that the single estimation of SOP is not reasonable, and the joint estimation in multiple states is inevitable.

## 3. Dual-State Estimation

The above analysis shows that the core states of the battery are coupled with each other. If any one of the three states is estimated separately without considering the other two states, the estimation accuracy will be limited; therefore, the joint estimation of the coupled states is necessary. From the summary analysis of the battery state estimation methods, it can be seen that battery SOC estimation is the basis for joint-state estimation. The accuracy of SOP and SOH estimation can be guaranteed only if the SOC estimation is accurate. For this reason, SOC/SOH and the SOC/SOP joint estimations are frequently performed when developing dual-state estimations [45–48]. Joint estimation enables each component to reach a higher estimation accuracy, thus achieving lower manufacturing cost, improved safety and power of electric vehicles, and wider promotion and popularization of electric vehicles [48].

Due to the different definitions, the joint estimation of SOC, SOH, and SOP could not be performed by the experimental method [49,50]. The data-driven methods build prediction models by constructing the mapping relationship between system inputs (current, voltage and temperature) and outputs, thus realizing joint SOC/SOH and SOC/SOP estimations based on data-driven methods [14]. Based on the fusion of the above two joint estimation methods, the joint estimation method for the numerical–modular hybrid class can be obtained. The methods for joint estimation are classified into three types: model-based, data-driven, and number-model fusion [51], which are shown in Figure 9.

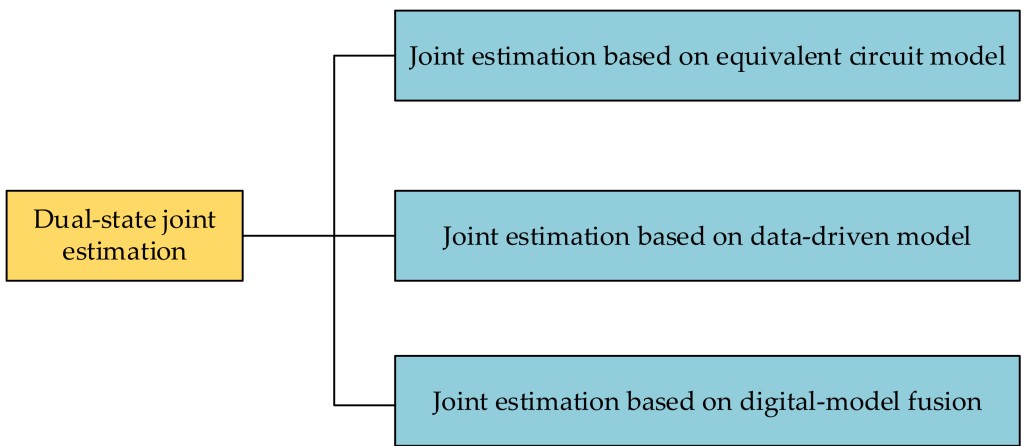

**Figure 9.** The classification of dual-state estimation methods.

### 3.1. SOC/SOH Joint Estimation

SOH is mainly influenced by SOC, temperature, discharge multiplier, cumulative battery life, number of charge/discharge cycles, and cumulative throughput of charge [5]. SOC is mainly influenced by temperature, battery health status, discharge multiplier, etc. The interactive relationship between SOH and SOC of the battery is shown in Figure 10.

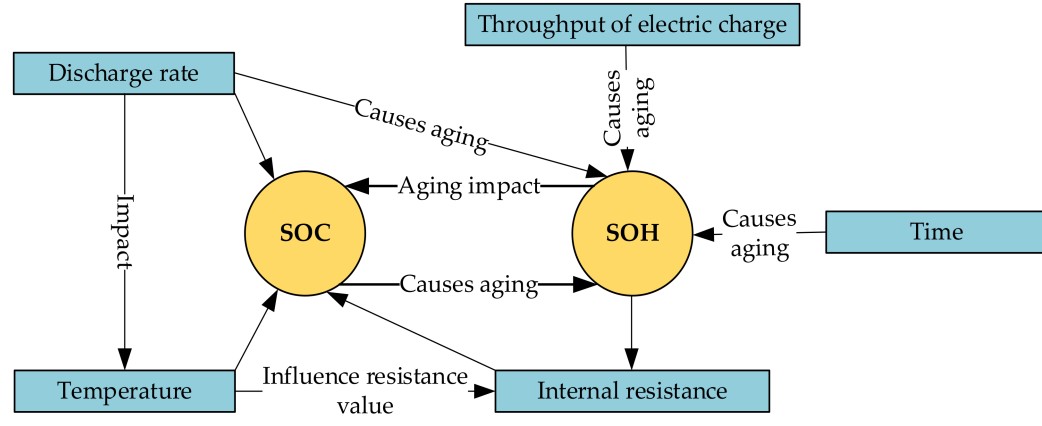

**Figure 10.** The interactive relationship between SOC and SOH.

The joint SOC/SOH-estimation methods are summarized in Figure 11.

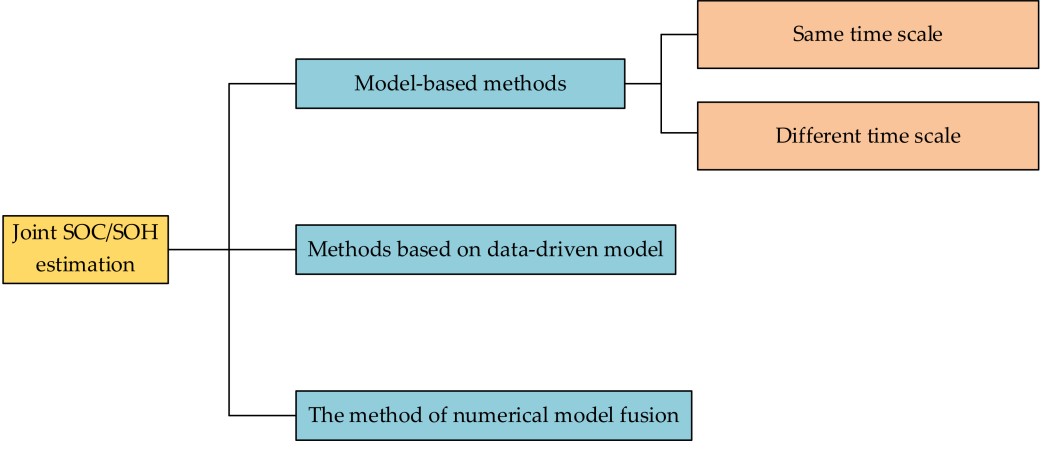

**Figure 11.** The joint SOC/SOH-estimation method.

### 3.1.1. Model-Based Estimation

Most of the relevant studies on joint model-based estimation use the equivalent circuit model (ECM), and, on top of that, various filtering algorithms are applied to estimate the battery state and aging parameters, respectively, which are coupled with each other as inputs. Since SOC and SOH have different time scales, two coupling ideas are derived from this: simultaneous estimation and joint multi-scale [19]. The joint estimation process at the same time scale is shown in Figure 12.

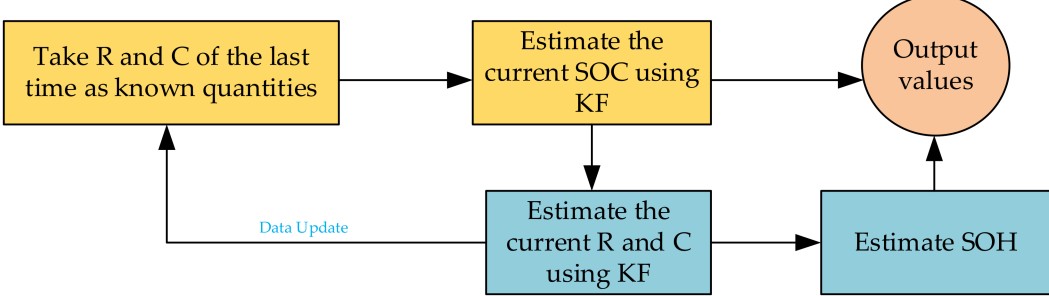

**Figure 12.** Joint estimation process at the same time scale.

Chunyang Zhao [49] has proposed the dual extended Kalman filter (DEKF) based on the ECM and selected ohmic internal resistance and capacity as one of the states. When SOC is used as a state for prediction, ohmic internal resistance and battery capacity are brought into EKF1 as known parameters for state prediction. When the ohmic internal resistance and capacity are the most-stated quantities for prediction, SOC is brought into EKF2 as a known parameter for state prediction to achieve iterative joint estimation, which improves the estimation accuracy. To solve the problem of the biased estimation of DEKF under strong nonlinear conditions, Prashant Shrivastava [9] proposed the dual unscented Kalman filter (DUKF), in which the ohmic internal resistance is considered as a state along with the capacity, and the two paths are alternated so as to achieve the joint estimation of SOC and SOH. To solve the uncertainty problem of noise during the driving of electric vehicles, Prashant Shrivastava [9] proposed the dual adaptive unscented Kalman filter (DAUKF) to realize the joint estimation of SOC and SOH on the basis of AUKF. AUKF1 and AUKF2 are chosen to calculate SOC and ohmic internal resistance, respectively, and the iterative calculation is continuously updated to realize joint estimation. Rui Zhu [51] has proposed the Multi-New Information Adaptive Untraceable Kalman Filter algorithm based on AUKF with the addition of multi-new information recognition theory. The battery capacity is used as a known parameter to achieve the real-time estimation of the SOC of the battery, and the SOC is used as a known quantity to estimate the capacity and SOH using a variable forgetting factor recursive least square algorithm. Hongyan Zuo [52] proposed to use a fractional-order model that can describe the battery performance more accurately. The SOC is estimated using the fractional order model-based extended Kalman filter (EKF), and the battery's internal resistance is estimated using the AUKF. The synergistic estimation of SOC and SOH is achieved by iteratively updating the internal resistance and SOC [53]. Hongyan Zuo used the multi-innovation adaptive unscented Kalman filter (MIAUKF) to estimate the SOC. Additionally, they used variable-forgetting factor recursive least squares (VFFRLS) to estimate the battery SOH. Then, the MIAUKF and VFFRLS were combined to realize the joint estimation of SOC and SOH. XinGao used the fraction order extended Kalman filter (FOEKF) to estimate SOC. The SOH was estimated by AUKF, and the internal resistance and SOC were iteratively updated to achieve accurate estimations of SOC and SOH. The above joint estimation is a simultaneous estimation, and the algorithm selection, model types, advantages, and disadvantages are summarized in Table 5. The estimation process for different time scales is shown in Figure 13.

**Table 5.** Joint SOC/SOH estimation at the same time scale.

| Algorithms | Model | Advantages | Disadvantages | Estimation Error |
|---|---|---|---|---|
| DEKF [53–55] | Thevenin model | High estimation accuracy. Faster convergence. Good robustness. | The effect of ambient temperature on the battery is not considered. Individual differences in batteries are not taken into account. Calculated losses exist. | <2% |
| DUKF [9,56,57] | 2RC model | Smaller error. Higher precision. Reflects actual battery characteristics. | The effect of ambient temperature on the battery is not considered, and the individual differences of the battery are not taken into account. | <2% |
| DAUKF [56,58] | 2RC model | Fast calculation speed. High estimation accuracy. Good convergence. | Not applicable to battery packs. Calculated losses exist. | <2% |
| MIAUKF + VFFRLS [59–61] | 2RC model | High accuracy and robustness. | The effect of temperature on the estimation accuracy is not considered. There is a computational loss. | <2% |
| FOEKF + AUKF [58,61,62] | 2RC fractional-order model | High precision. High self-adaptability. Fast convergence speed. | Highly influenced by temperature. Not applicable with battery packs. High calculation volume. | <1% |

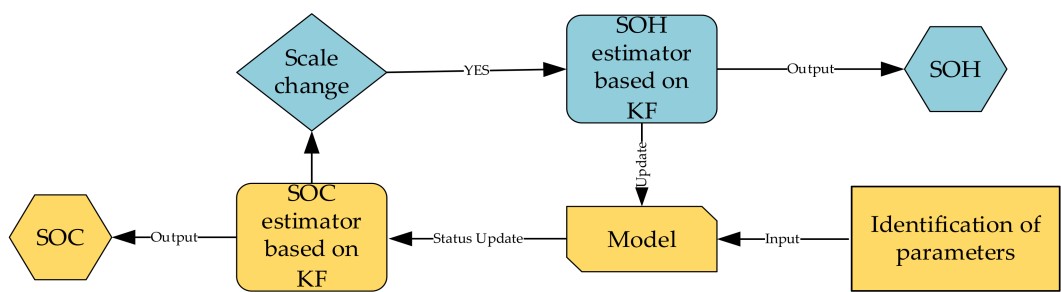

**Figure 13.** Joint estimation process at different time scales.

Yuan Zou [63] proposed to establish multi-time scale state space equations, construct multi-dimensional spatial interpolation surfaces of SOC/SOH and battery model parameters, and realize the joint estimation of SOC and SOH based on a unified particle filter (UPF). Updating the capacity parameters and model parameters in the SOC estimation based on the current SOH estimates improves the long-term estimation performance of the SOC. Online joint estimation is achieved using the online health indicator (HI) as the measurement value, instead of the battery capacity in the SOH estimation. He Yao [54] introduces macroscale and microscale criteria L and k. A time update and measurement update are performed when the microscale criteria accumulate to the macroscale criteria L [64–66]. The macroscopic time scale uses the EKF algorithm for parameter (capacity) estimation, which leads to the SOH value. At the microscopic time scale, the Adaptive square root extended Kalman filter (ASREKF) algorithm is used for charge-state SOC estimation [67,68]. Ehprem

Chemali [67] proposed to increase the parameter identification scale in SOC estimation and use the recursive least squares method with a forgetting factor to estimate the cell parameters to obtain the ohmic internal resistance and the SOH, while updating the parameters to estimate the SOC. Ruxiu Zhao [68] proposed to use EKF1 for microscopic SOC estimation and EKF2 for macroscopic-rated capacity estimation, thus realizing the joint estimation of the two states. ShaoDong Cui [69] proposed a multi-timescale EKF using two observers at once at the macroscopic scale to estimate SOH and at the microscopic scale to estimate SOC. The advantages and disadvantages of the above SOC/SOH joint estimation methods at multiple time scales are summarized in Table 6.

**Table 6.** Joint multi-timescale SOC/SOH-estimation methods.

| Algorithms | Model | Scales | Disadvantages | Estimation Error |
|---|---|---|---|---|
| UPF + UPF [63,64] | 1RC model | SOH estimation scale is one charge/discharge cycle, SOC estimation interval is 0.1 s. | Higher precision. Lower computational volume. Low hardware requirements. | <1.5% |
| ASREKF + EKF [64,65] | Thevenin model | SOH estimation scale is one charge/discharge cycle, SOC estimation interval is 1 s. | High accuracy and low calculation volume. | <1.5% |
| FFRLS + DEKF [68–70] | 2RC model | SOH estimation scale is 2.5 s, SOC estimation interval is 1 s. | High stability and accuracy, saving calculation cost. | <1.5% |

The joint estimation of multiple time scales reduces the computational cost while ensuring the estimation accuracy, with the disadvantage that the effects of ambient temperature and the number of cycles are not considered. The joint model-based SOC/SOH estimation relies on the accuracy of the model, when the battery is not modeled accurately, the estimation accuracy cannot be guaranteed. A complex model is better able to describe the internal dynamic of the battery, which not only improves the estimation accuracy, but also results in a higher computational burden [71]. How to balance the relationship between model and estimation accuracy and computational speed is a difficult problem that has not yet been solved for joint estimation based on models.

### 3.1.2. Data-Driven Estimation

In order to avoid the effects of the insufficient accuracy of the ECM, a data-driven joint estimation method is proposed. In the data-driven method, the mapping relationship between inputs (voltage, current, temperature, or internal resistance) and SOC and SOH is constructed by date-driven model [2]. Taking the SOH estimation value into account in the SOC estimation can eliminate the negative impact of the aging factor of a Li-ion battery [17,22,72–74]. The flow of the data-driven approach is shown in Figure 14. Table 7 summarizes the use of a data-driven method for joint estimation methods in recent years.

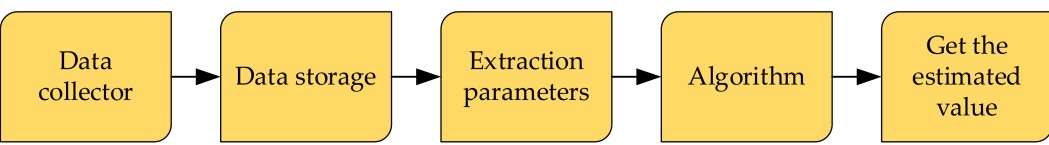

**Figure 14.** The process of data-driven approach.

**Table 7.** Joint data-driven SOC/SOH-based estimation method.

| Algorithms | Data Set | Advantages | Disadvantages | Estimation Error |
|---|---|---|---|---|
| Double Recurrent Neural Network [17,72,73] | NASA Lithium Battery Random Use Dataset | Independence of battery model. No decoupling required. | Need to keep updating data. | <1% |
| LSTM [74] | Oxford Aging Dataset | Accuracy is higher than wavelet neural network and BP neural network, and the long-term dependency problem is solved. | No validation of effectiveness on battery packs. | <1% |
| SWPSO-DRNN [22,75,76] | Customized data sets | Higher training effect than gradient descent algorithm. High generalization ability and robustness. | Not applicable to battery packs. | <1% |
| GRN-RNN + CNN [77–79] | NASA Lithium Battery Random Use Dataset and Oxford Battery Aging Dataset | Avoid long-term dependence on new information, fast computation, high accuracy, and good robustness. | No validation of effectiveness on battery packs. | <1% |
| Mogrifier LSTM-CNN [79] | NASA dataset and Oxford aging dataset | Solve the problem of large local errors of LSTM method, good adaptability and high robustness. | No validation of effectiveness on battery packs. | <1% |

The data-driven joint estimation obtains the SOC and SOH variation characteristics by machine learning without relying on the accurate model or decoupling [17].

Shuo Li [17] selected the historical operating data of the battery voltage, current, surface temperature, SOC, and SOH as feature quantities and applies long short-term memory (LSTM) to predict the voltage and surface temperature in subsequent cycles. Based on the predicted voltage and temperature, the LSTM deep learning network is used to jointly predict the trends of SOC and SOH [74]. Hicham Chaoui [73] proposed to build a double-recurrent neural network (DRNN) with a dynamic mapping capability based on a nonlinear autoregressive exogenous (NARX) structure for battery SOC and SOH estimation, introduced the self-adaptive weight particle swarm optimization (SWPSO) algorithm to train the DRNN on the basis of appropriately selected training data and test data, and compared it with the gradient descent training method. The results show that SWPSO-DRNN can effectively compensate for the influence of temperature and aging and improve estimation accuracy. Marcantonin Catelani [44] applied a recurrent neural network with a gated recurrent unit for SOC estimation and a convolutional neural network for SOH estimation. Based on the voltage, current, and temperature acquired from BMS, the convolutional neural network (CNN) is capable of SOH estimation. Jinpeng Tian [37] proposed that both SOC and SOH estimation are based on RNN estimation, the gated recurrent unit and recurrent neural network (GRU-RNN) used in the joint estimation method for SOC estimation is able to use historical information far from the current state, avoiding the long-term dependency problem in RNN, and has a stronger data-feature-extraction capability and higher accuracy. The CNN used for SOH estimation has the advantages of fewer parameters, lower computational burden, and a smaller memory footprint [76,77]. The Mogrifier LSTM neural network has a stronger nonlinear mapping capability than the original LSTM neural network, a higher accuracy of SOC estimation,

and different estimation scales for SOC and SOH estimation, which can greatly reduce the computational effort of the joint estimation model [78].

The data-driven method has a higher estimation accuracy compared with the model method, but the collection of training data is time-consuming, and the quality of training data directly affects the estimation accuracy. To solve the above problem, the method of data-model fusion is proposed for joint estimation, which reduces the computational effort of using a data-driven approach alone, while obtaining a higher accuracy than that of estimations using models alone.

### 3.1.3. Fusion Estimation Algorithm Based on Data-Driven Approach and Model

The complex electrochemical mechanisms and highly nonlinear coupling of the parameters of the battery have led to a number of unresolved or further developed problems in lithium-ion modeling and battery-state estimation: scarcity of coupled modeling studies of temperature, electrical, and attenuation behavior [14,31,34]. Generally, model-based joint estimation is highly dependent on model accuracy, while data-driven estimation is computationally expensive, relying on training data. To conquer the defects of both methods, a joint estimation method based on data-driven and model-fusion approaches is introduced.

Consider the mutual-influence relationship between SOC and SOH: SOH determines the capacity in the SOC equation on large time scales, while, on small time scales, SOC affects the rate of change of SOH. Considering the interaction between SOC and SOH, the coupling relationship is decoupled, two estimation algorithms are used to update SOC and SOH separately, and the two estimators update each other's states and parameters internally. The remaining battery power can be obtained directly from the relationship with current and voltage, and the model is easy to build and robust [64], so a model-based approach is considered to estimate the SOC. The battery-aging model is difficult to establish and needs to explore the pattern from a large amount of data, which can be solved more easily and quickly by adopting a data-based approach.

Yuan Zou [63] has proposed a fusion method to construct a coupled thermal–electrical-aging cell model for SOC/SOH joint estimation. In the estimation model, UKF is introduced to estimate SOC, LSTM-RNN to estimate SOH, and the UKF-NN method with dual time scales is designed to estimate SOC and SOH jointly. Figure 15 illustrates the flowchart of the online implementation of this joint estimator.

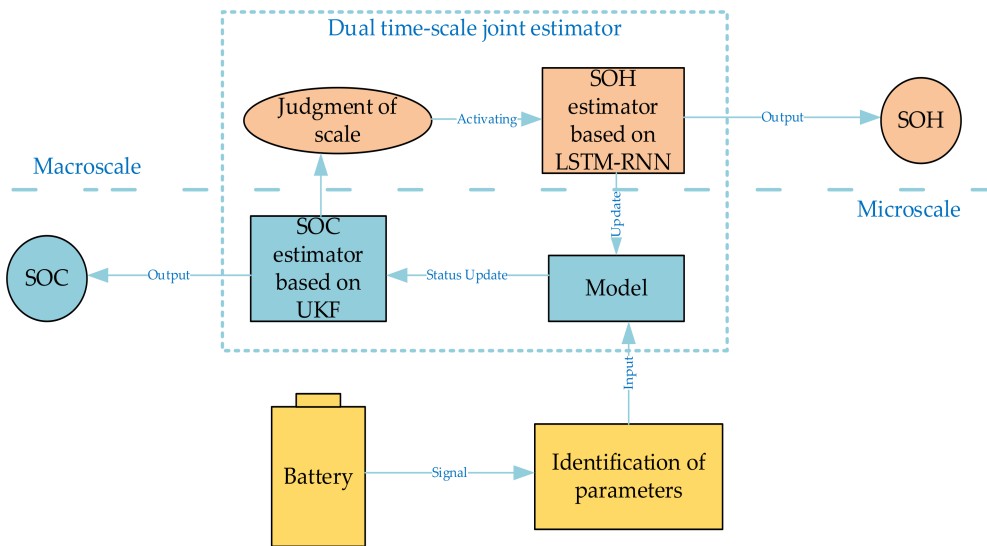

**Figure 15.** Flow chart for joint estimation of dual time scale.

Arpita Mondal [78] proposed a joint battery multi-timescale state-estimation method based on the data-driven method (DDM) and ECM. The SOC is estimated at the microscopic

scale and with microscopic cell parameters, while the SOH is estimated at the macroscopic scale [79]. The overall flow chart is shown in Figure 16.

**Figure 16.** Block diagram of joint SOC/SOH estimation.

Step 1: For the cycle period before the prediction's starting point (SP), establish the Thevenin model, use the LS algorithm to identify the resistance–capacity parameters, extract the internal resistance as heart failure (HF), and establish the LSSVM battery-aging model.

Step 2: For the Nth cycle (N > SP), identify the internal resistance value under the current cycle as HF and input it into the LSSVM aging model for SOH estimation.

Step 3: Select the system state variables as SOC and polarization capacitance; the input variable is current, and the output variable is terminal voltage. Identify the value of resistance and capacitance, and update it in each cycle, followed by recursive operations using the UKF algorithm to achieve SOC estimation.

The number-model fusion method integrates the advantages of model-based methods and data-driven methods [80,81], thus reducing the dependence of estimation accuracy on model accuracy and the computational cost. The above advantages make the number-model method fusion an effective method for battery joint estimation.

*3.2. SOC/SOP Joint Estimation*

SOC and SOP are the key pieces of information for the vehicle control strategy, which assists the EMS to realize power distribution [82–85]. The relationship between SOC and SOP is shown in Figure 17. SOC is a constraint for SOP estimation, and SOP indirectly affects SOC estimation by influencing the discharge C-rate and cell resistance. Since SOC is the constraint quantity for estimating SOP, most scholars use ECM to estimate SOC and then perform the joint estimation of SOC and SOP.

Theoretically, the more RC networks that are connected in series in the multi-order model, the higher the accuracy of the battery model. However, in practical application, the terminal voltage response of the RC network with a second order or above is not much different and is close to the terminal voltage response of an actual battery. The increase of the order will greatly increase the number of calculations completed by the processor, which is obviously not worth comparing with the slight improvement of the accuracy. Therefore, in actual use, in order to give consideration to model accuracy and calculation complexity, the 2RC equivalent circuit model is generally used to estimate SOP. Chunyang Zhao [49] proposed a detailed method for estimating peak power based on ECM. Improved first-order ECM is proposed, the recursive extended least squares (RELS) method is introduced for parameter identification, and SOP is estimated based on voltage and

current constraints in SOC/SOP joint estimation. As the model's accuracy is not sufficient, and the constraints for SOP estimation are incomplete, joint SOC/SOP estimation with multi-parameter constraints for the second-order ECM was performed. The constraints for battery SOP mainly include voltage, current, SOC, and temperature, and most studies select voltage, current, and SOC constraints or temperature, voltage, current, and SOC constraints [86–88]. The SOP-estimation methods based on the second-order equivalent model under voltage, current, and SOC constraints are summarized in Table 8.

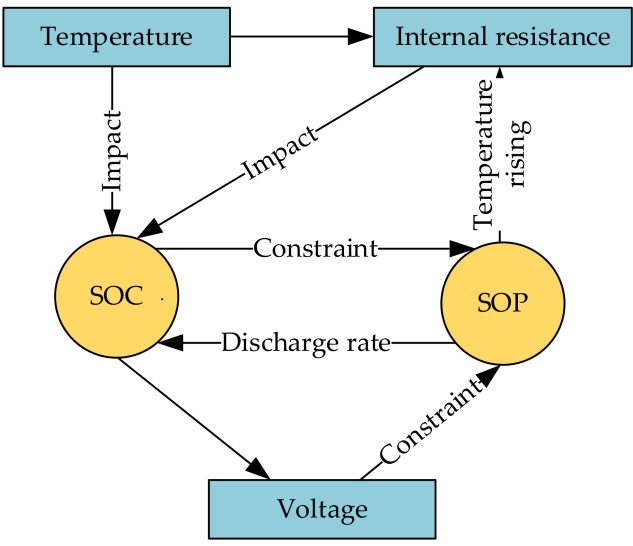

**Figure 17.** The relationship between SOC and SOP.

**Table 8.** SOP-estimation methods based on 2RC model with multiple constraints.

| Model | Estimation Method of SOC | Constraints | Advantages | Disadvantages | Estimation Error |
|---|---|---|---|---|---|
| 2RC model | EKF | Voltage, current, SOC | The estimation accuracy is higher and more robust than that without considering SOC constraints. | No consideration of temperature and aging effects. | <5% |
| 2RC model | H infinity filter | Voltage, current, SOC | Better robustness and adaptability than EKF. | No consideration of temperature and aging effects. | <2.5% |
| 2RC model | UKF | Voltage, current, SOC | High estimation accuracy and good robustness. | No consideration of temperature and aging effects | <2% |
| 2RC fractional-order model | Fractional-order adaptive extended Kalman filter (FO-AEKF) | Voltage, current, SOC | Better robustness and adaptability than EKF. | No consideration of temperature and aging effects. | <3% |
| 2RC fractional-order model | Square-root unscented Kalman filter (SRUKF) | Voltage, current, SOC | High estimation accuracy and good robustness. | No consideration of temperature and aging effects. | <2% |

The accuracy of SOP estimation is improved when second-order ECM is applied or when voltage, current, and SOC are used as constraints. However, the variation of temperature or parameters (internal resistance, capacity) are often ignored, which has significant influence on the accuracy of the SOP estimation. For example, the rising of temperature will accelerate the occurrence of side reactions inside the lithium battery [47,89];

the decrease in temperature will cause the deposition of active lithium on the electrode surface, and all these cause changes in the usable capacity, internal resistance, and other characteristic parameters of lithium batteries [8–10]. As a result, it is essential to include temperature as a SOP constraint to achieve more accurate SOP estimation.

Among existing SOP-estimation methods, the common type of method of considering temperature is building the relationship between temperature and battery parameters to estimate SOC through parameter variation and then selecting SOC as a constraint to achieve the temperature-to-SOP estimation constraint [90–92]. The second type of method is to obtain the maximum thermal efficiency under the temperature constraint and then adjust the current constraint to achieve the temperature constraint on SOP [38,93,94]. The first type of method is represented by Yuanwang Deng [40], whose estimation process is shown in Figure 18.

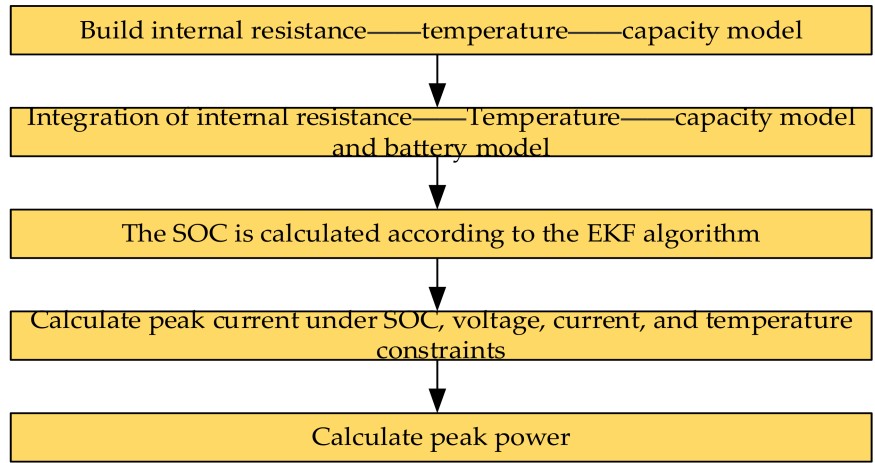

**Figure 18.** The estimation process of the first type of method.

Step 1: Measure the relationship between the capacity and internal resistance of Li-ion battery at different temperatures and use the Arrhenius equation to fit a curve to the relationship between them. The capacity temperature compensation coefficient and internal resistance temperature compensation coefficient are obtained.

Step 2: Bring the two compensation coefficients into the state and observation equations that characterize the SOC estimation process.

Step 3: Discretize the state equation with compensation coefficients and the observation equation, and obtain the current SOC value of the battery using the EKF algorithm.

Step 4: Calculate the peak current under SOC, voltage, and current constraints.

Step 5: Calculate the peak power under multiple constraints by the peak current.

The above steps achieve the temperature constraint on peak power, improve the temperature adaptation of peak power estimation, and increase the constraint to obtain a higher estimation accuracy with stronger robustness and safety of battery use.

The second type of method is represented by Marcantonin Catelani [44]. The implementation flow is shown in Figure 19.

Step 1: Establish the thermal model structure of the battery and calculate the expression of the heat generation rate according to the law of energy conservation. Calculate the relationships between the birth heat rate and current, cell temperature, ambient temperature, thermal resistance, thermal capacity, and thermal time constant.

Step 2: The temperature rise curves of the constant current discharge phase at different temperatures are used. Use a genetic algorithm to obtain the values of thermal resistance and thermal capacity at different ambient temperatures.

Step 3: Calculate the maximum heat-generation rate of the battery under this constraint using the maximum battery temperature as a constraint.

Step 4: Substitute the maximum heat-generation rate into the heat-generation rate expression to obtain the current under the temperature constraint.

Step 5: Estimate the battery SOC using the extended Kalman filtering method.

Step 6: Use the dichotomous method to find the peak current under the joint constraint of SOC, terminal voltage, temperature, and cell current.

Step 7: Calculate the peak power of charging and discharging Li-ion batteries under multiple constraints. The engineering uses the temperature rise rate of the battery as one of the constraints to estimate the SOP with high accuracy and robustness, which improves the safety of the battery.

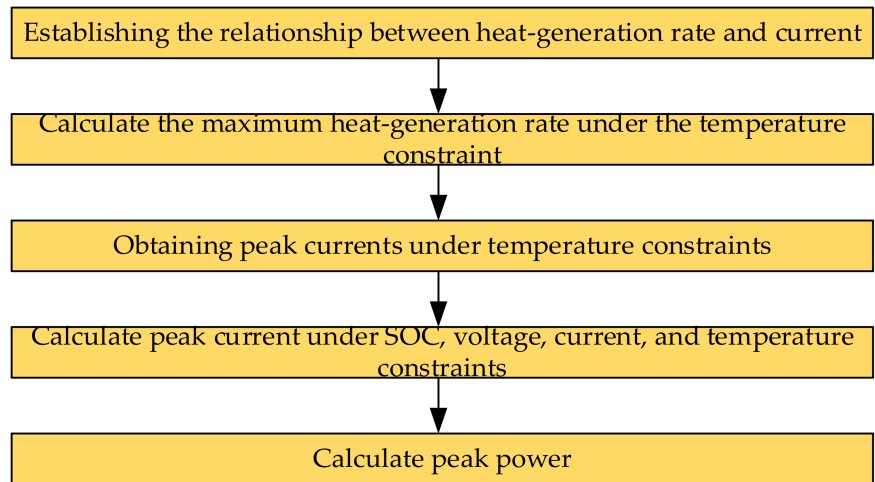

**Figure 19.** The estimation process of the second type.

## 4. Key Issues and Future Work

Based on the description above, the two-state joint estimation approach has been well developed. However, there is still much room for improvement in terms of estimation accuracy and computational efficiency for online applications. SOC, SOH, and SOP, as a dynamically coupled system, are subject to many factors in practice. On this basis, the difficulties to be solved in the joint estimation of SOC, SOH, and SOP are analyzed from three aspects, and the outlook on the future development of joint estimation is proposed from three aspects, as shown in Figure 20.

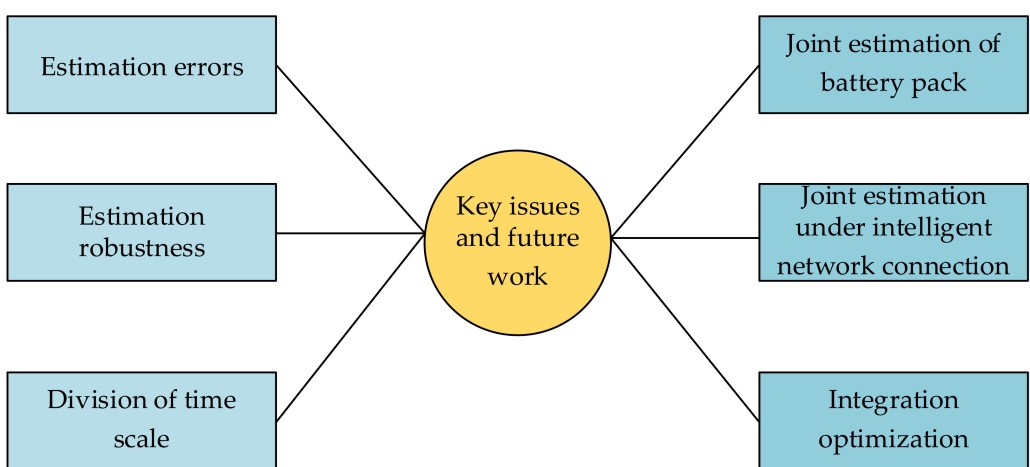

**Figure 20.** Key issues and future work.

*4.1. Key Issues*

4.1.1. Estimation Errors

Battery-state estimation error come mainly from battery modeling, data collection, estimation algorithms design, etc. The accumulation of these errors may eventually result in significant estimation errors. First, no model can fully represent the nonlinear behavior of lithium batteries. For example, hysteresis effects increase the uncertainty of modeling. Therefore, a more accurate model is needed to represent cell properties by combining thermal, electrochemical, and series-parallel circuit models. Second, the inaccuracy of model-parameter identification affects the accuracy of state estimation. Third, data measurement noises from current, voltage, temperature, and other sensors are inevitable. The accumulation of errors can lead to large errors in the state estimation of the battery, so it is of high importance to eliminate these errors. By improving the experimental conditions, measurement errors can be eliminated to some extent. Finally, as mentioned before, the estimation process also introduces an online application process and measurement noise. Therefore, improving existing estimation methods is essential for reducing systematic errors and achieving accurate SOC, SOH, and SOP estimations.

4.1.2. Estimation Robustness

For single-state estimations of SOC, SOH, or SOP, the robustness of the estimation method can be easily guaranteed, as long as the development process is correct. and the algorithm parameter is adjusted appropriately. However, when designing the estimation method for coupled states of the battery, it is difficult to assure the robustness of the algorithm because different methods are applied to each state, and it is hard to achieve combined regulation of methods. As a result, coupling errors may appear in the joint SOC/SOH estimation; when there is a large error in the SOC estimation, a reversal error could generate in the SOH estimation, thus leading to simultaneous divergence in the SOC/SOH estimation, while the total error of the joint estimation is inside of the permitted range. In addition, as the definition and the calculation of SOC/SOH/SOP at the pack level are totally different, there is a lack of experimental verification of whether the joint estimation method proposed for the single cell level is robust when applied to the battery pack SOC/SOH/SOP estimation.

4.1.3. Division of Time Scale

In terms of battery parameters, SOH changes slowly with time, the SOC estimation iteration time is millisecond level, and continuous SOP iteration time is second level. Using the same time scale does not yield accurate and reliable estimates and greatly increases the computational effort of the control system and reduces stability. Incorrect time scale division can cause the transmission of errors, making SOC, SOH, and SOP disperse at the same times. For this reason, it is the future direction of joint estimation to reasonably divide the scale of each state estimation in the joint estimation process.

*4.2. Future Work*

4.2.1. Joint Estimation of Battery Pack

For lithium batteries, performance may vary greatly for different manufacturers or batches. Even for the same branch and batch, there may be differences in the dynamic characteristics among cells, which will gradually increase with time.

As a result, huge differences exist in estimating the SOC/SOH/SOP between the cell and pack. When developing a joint estimation method for the battery pack, more complicating factors have to be considered, such as inconsistency in cell capacity, resistance, temperature, and SOC; and the joint estimation algorithm structure designation against cell-module-pack states coupling mechanism; and the algorithm optimization for the reduction of the central processing unit (CPU) computational burden. In the future, an 800 V pack system will become mainstream, and packs will become more and more complex; the

research into SOC/SOH/SOP estimation at the pack level will be the key issue affecting BMS performance.

### 4.2.2. Joint Estimation under Intelligent Network Connection

SOH measurement in the full life stage of the battery requires a lot of time and computational cost, for which data-driven SOH estimations can be established on the Cloud Platform. The SOH and parameters of different vehicles are uploaded to the Cloud Platform and categorized and stored, and the battery is monitored, predicted, and simulated on the Cloud Platform. When an electric vehicle is started, the initial parameters are matched by the cloud for joint estimation based on the state at start-up, and the data are continuously uploaded during driving engineering. The computing unit exists independently in the cloud to reduce the computation burden and free up storage space in the vehicle, thus improving the whole operation performance of the BMS.

### 4.2.3. Integration Optimization

In the BMS, the calculation modules of SOC, SOH, and SOP are separated from each other, and multiple modules are required to complete a core-state calculation. This increases the computational burden of the control system. To this end, a hypothesis is proposed: the modules of SOC, SOH, and SOP are integrated and synthesized into a single module. In this way, the memory requirement for computing is reduced, and lag caused by information transfer between multiple modules is avoided, thus cutting BMS costs.

## 5. Conclusions

This paper reviews the development trends of SOC, SOH, and SOP estimation techniques for power batteries and summarizes the advantages and disadvantages of estimation methods.

After this it analyzes the coupling relationship among SOC, SOH, and SOP, points out the defects of single-state estimation, and reviews the development of joint estimation of SOC/SOH and SOC/SOP. For the problem of joint SOC/SOH estimation, model-based, data-driven, and number-model fusion estimation methods are summed up. For the problem of joint SOC/SOP estimation, a joint estimation method based on two types of constraints is proposed. The methods and joint mechanisms used for the joint estimation of SOC/SOH and SOC/SOP are described in detail. These methods are discussed and evaluated, and their advantages and disadvantages are summarized.

Based on the shortcomings of single-state and dual-state estimation methods, this paper summarized three unresolved issues: estimation errors, estimation robustness, and the division of time scale. These three unresolved issues will be the focus of future research. This paper raised three outlooks for the future of the industry: joint estimation of battery pack, joint estimation under an intelligent network connection, and integration optimization. These three perspectives offer new possibilities for better development of BMS.

**Author Contributions:** J.H. put forward key issues and future work of the joint estimation methods; T.L. reviewed the single-state estimation methods; F.Z. improved the overall structure of the paper and edited the paper; D.Z. and Y.Z. reviewed joint estimation methods; L.Y. provided key issues in SOH estimation; L.Z. provided all tables and figures. All authors have read and agreed to the published version of the manuscript.

**Funding:** This research was funded by Key Scientific and Technological Project of Henan Province, grant number: 222102220106, 212102210237 and 222102240053; Key Scientific Research Projects of Institutions of Higher Learning in Henan Province, grant number: 22A460033.

**Institutional Review Board Statement:** Not applicable.

**Informed Consent Statement:** Not applicable.

**Data Availability Statement:** Not applicable.

**Conflicts of Interest:** The authors declare no conflict of interest.

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
