# Peer review of "A Review of Critical State Joint Estimation Methods of Lithium-Ion Batteries in Electric Vehicles"

_wevj, doi:10.3390/wevj13090159_

Round 1

Reviewer 1 Report

1The SOCSOH and SOP are not clearly defined, which are important for the estimation methods development.

2In Table 456 and 7, some abbreviations have not been defined, please give definition when abbreviations appear in the paper for the first time.

3In Chapter 3.1.2 the data-driven estimation method, as the SOC of each cell in the battery pack must be estimated, how is the feasibility of data-driven methods when implemented in BMS?

4In Table 7, all the SOP estimation methods select 2RC model, why other ECMs are not listed, such as 1RC model or Thevenin model?

5In Chapter 4.2.2 the joint estimation under intelligent network connection, how to use intelligent network connection technology to estimate states for different type of battery packs or vehicles simultaneously?

Reviewer 2 Report

The authors classified different methods of single-state and two-state joint estimation of lithium-ion battery, and gave a future work guidance. However, there are still some problem need to be addressed.

1.       For some novel or important method, you are supposed to add more of detailed explanation and figure.

2.       Figure 2 is not intuitive enough for some layout reason, you’d better make a change.

3.       More of accurate indicator information are supposed to be given in tables instead of indistinct description.

4.       Instead of using word KF1 and KF2 in figure 6, which is untidy, you can just use KF and explain in the following that the two KF are different.

5.       Except concluding the work you have done in the conclusions part, it is recommended to add some your own view.

Reviewer 3 Report

The paper is well organized ad well presented.

Many review papers on SoC and SoH estimation exists, the paper should add (in the beginning or at the end) a list of review papers and the key differences and improvements of this work with respect to other review.

Reviewer 4 Report

To Authors,

Concerning the manuscript entitled “A Review of Critical States Joint Estimation Methods of Lithium-ion Batteries in Electric Vehicle” (ID: wevj-1846495) by Junjian Hou, Tong Li, Fang Zhou, Dengfeng Zhao, Yudong Zhong, Lei Yao, Li Zeng:

Q1:  in line 46, some sentence is repeated.

Q2: in line 65, the term “manipulation” is ambiguous.

Q3: in lines 119-124, the authors refer to “Hybrid-driven approach is proposed to combine the rapidity of model method estimation…” to conclude with “… disadvantages such as high computational complexity and slow estimation speed”. This is counter-intuitive.

Q4: the paragraph in lines 263-269 is not clear. SOH updates over hours. SOC updates continuously every millisecond. How is this leading to instability in the SOC estimates? Perhaps this paragraph should be discarded. The information is clear in the following one, including the information in Table 5.

Q5: not all algorithms in Tables 4-6 are defined (e.g., like “Adaptive Square Root Extended Kalman Filter” for ASREKF in Table 5). The authors can define the acronyms (including the references) in the table caption.

Q6: Funding and Acknowledgments contain the same information. 

Round 2

Reviewer 2 Report

No more comments